



**Positive tipping points for accelerating adoption of regenerative practices in African**
**smallholder farming systems: What sustains adoption?**
Authors: Antony Philip Emenyu[1*], Thomas Pienkowski[2], Andrew M. Cunliffe[1], Timothy M.
Lenton[1], Tom Powell[1]
[1] Global Systems Institute, Faculty of Environment, Science and Economy, University of Exeter
[2] Centre for Environmental Policy, Faculty of Natural Sciences, Imperial College London
Antony Philip Emenyu: ae474@exeter.ac.uk
Tom Powell:  t.powell@exeter.ac.uk



**Abstract**
Regenerative agriculture (RA) practices have been promoted as a critical climate change resilience
strategy and adaptation solution for smallholder farmers in Sub-Saharan Africa. However, most RA
programmes struggle with securing and sustaining high adoption rates with many facing dis-adoption.
We used Lenton et al.'s positive tipping points framework to assess the potential for fast and lasting
adoption of Regenerative Agriculture (RA) in Sub-Saharan Africa. This involved reviewing literature
and combining evidence from the successful expansion of the International Small Group and Tree
Planting Program (TIST) in East Africa to examine the conditions and feedback processes that drive
RA adoption. We found that the key leverage points for TIST wide and rapid adoption were: (1) the
cultivation of reinforcing feedback processes that strengthened the social capital around adoption and
(2) elimination of barriers to carbon accreditation. Integrating carbon accreditation protocols as
standard in design or review of RA interventions could provide an essential leverage to boost adoption
rates. Future studies could explore what drives variations in scaling rates and patterns between the
sites to inform more site specific interventions.
Keywords: International Small group and Tree Planting programme (TIST), agroforestry, reinforcing
feedback, climate change resilience
**1.0 Introduction**
Smallholder farms account for close to 80% of all farms in sub-Saharan Africa (OECD-FAO, 2016)
and are often characterised by rainfed farming on highly degraded soils, where farmers have limited
capital resources to invest in improving their production systems. These characteristics make
smallholder farmers highly vulnerable to effects of climate change, placing them at a high risk of food
and livelihood insecurity. The Intergovernmental Panel on Climate Change (IPCC) (2022) Working
Group II report states that most smallholder farmers in Africa and the global south have already
reached their soft limits for human adaptation. Implying that, while certain adaptation options could
exist, they remain inaccessible to smallholder farmers due to financial, governance, institutional and
policy constraints. At the same time, the impacts of climate change are worsening across Africa. For
instance, under the current emissions trajectory, Coupled Model Intercomparison Project Phase 5
estimated that temperatures across Africa would increase by 2.7°C by 2050s while rainy seasons
would shorten, accompanied by more intense rain events (Girvetz et al., 2019). Such changes could
result in irreversible losses in productivity, and potentially the complete collapse of current
agricultural production systems, leading to high food insecurity. The latter risk is amplified by the
limited ability of smallholders to adapt.
In recent years, regenerative agriculture (RA) has gained traction in policymaking. Both the Sharm
El-Sheikh Adaptation Agenda and the Breakthrough Agenda recognising the need for a mass
transition to RA by 2030 to strengthen the resilience and adaptability of smallholder farmers to the
impacts of climate change (FOLU, 2021; Marrakech Partnership for Global Climate action, 2022). RA
here refers to farming practices that improve soil, water and overall ecosystem health, increase carbon
sequestration, increase biodiversity, maintain or improve farm productivity and improve social and
economic wellbeing (see Newton et al., 2020). Such practices could include conservation agriculture,
agroforestry, and permaculture. According to the International Union for Conservation of
Nature(IUCN, 2021), with just 50% adoption of RA, African smallholder farmers could potentially
see a 30% reduction in soil erosion, up to a 60% increase in water infiltration rates (reducing run-off
and increasing soil water storage), a 24% increase in nitrogen content and at least a 20% increase in
soil carbon content. This could add approximately $70bn gross value per year to African farmers
(IUCN, 2021). However, despite the evidence of the various benefits of RA, programmes promoting
RA across the continent have struggled to quickly attain and sustain scale. While several studies look
into factors that influence adoption of various RA practices across the continent (see Bouwman et al.,
2021; Grabowski et al., 2016; Guteta & Abegaz, 2016), there is still little understanding of what could





enable rapid scaling. As a result, most RA programmes, despite managing to secure some early
adoption success, fail to reach adoption tipping points, instead stagnating or experiencing dis-adoption
(Grabowski et al., 2016; Habanyati et al., 2020; Kehinde & Adeyemo, 2017). Without an
understanding of processes driving rapid transition from initial early adoption success to continuously
higher and sustained adoption rates, most RA programmes will continue to struggle to attain scale.
Lenton *et al.* (2022) advanced the idea that some actions can trigger or strengthen reinforcing
feedback processes that in turn drive rapid adoption of interventions in social-technological-ecological
systems. This reasoning was brought together in a conceptual framework for operationalising Positive
Tipping Points (PTPf), which identifies typologies of reinforcing feedbacks and enabling conditions
that can trigger positive tipping points, and interventions that could accelerate them. A corresponding
report (FOLU, 2021) proposed that these dynamics could be occurring for farmers in parts of India
but this has not been rigorously assessed in African farming systems.
In this paper, we build an understanding of the enabling conditions and reinforcing feedback
processes for accelerated and sustained adoption of RA to help inform efforts to rapidly scale these
RA strategies as an urgent response to the climate change pressures on smallholder farming systems.
We first review literature on adoption of various RA practices such as conservation agriculture,
agroforestry, and climate smart agriculture to identify various enabling conditions that seem to favour
or discourage adoption. We then focus on The International Small group and Tree planting programme
(TIST) in East Africa as a case study to illustrate how the various enabling conditions and reinforcing
feedback processes function in a practical context. Finally, we explore what lessons could be drawn
from the scaling of TIST to develop an understanding of potential leverages to trigger accelerated
adoption of RA in Africa. In the next section we provide a brief RA focused introduction of the PTPf.
After this we introduce how TIST applies various aspects of this framework and finally discuss what
lessons can be drawn from the data on TIST to inform other programmes seeking to adopt this
approach.
The PTPf identifies various enabling conditions, reinforcing feedback processes and possible
interventions that could ignite system level transitions towards a positive tipping point (see Figure 1).
See Ong et al.(2023) for an illustration of how some of these tipping points dynamics could operate in
real world systems such as a packaging system.



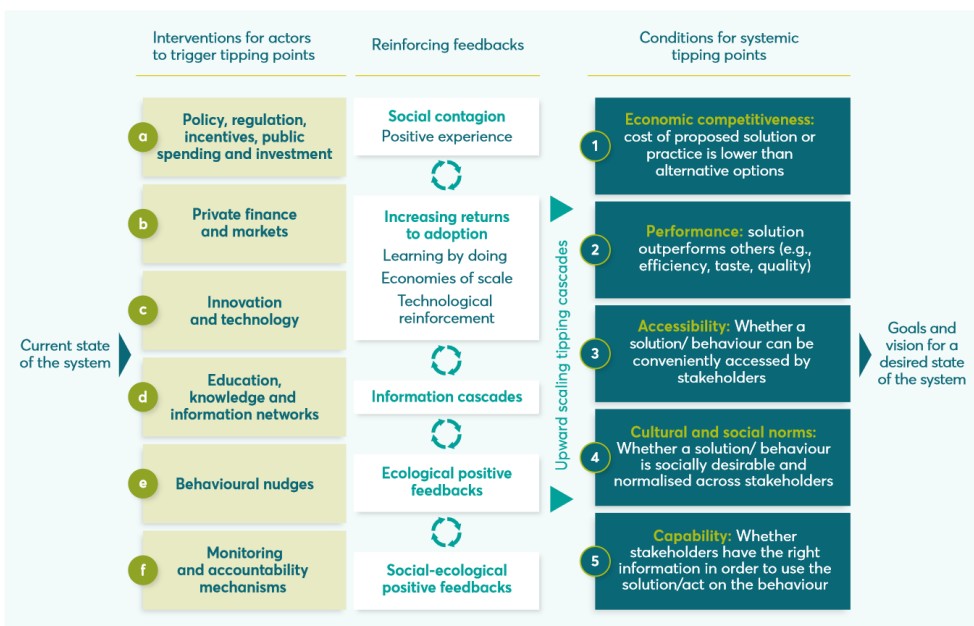


*Figure 1: Framework for operationalising positive tipping points adopted from the Food and Land Use Coalition (FOLU) report on accelerating the 10 critical transitions (FOLU, 2021, p. 7).*

## 2.0 Enabling conditions for successful adoption of RA in Africa

Adoption is a complex process with multiple possible outcomes; adoption (continued application of the practice) (Ainembabazi & Mugisha, 2014; Amadu et al., 2020), partial adoption (applying part of the practice) (Zulu-Mbata et al., 2016), changes in adoption intensity (applying more or less of the practice) (Kunzekweguta et al., 2017; Mujeyi et al., 2022), non-adoption (not-applying the practice) (Khoza et al., 2019), dis-adoption (stopping application of the practice) (Alpizar et al., 2022; Grabowski et al., 2016), and adaptation (editing the practice) (Bouwman et al., 2021). Several key factors increase the likelihood of successful adoption: the intervention has to be economically competitive, culturally and socially appropriate, easily accessible and outperform other alternatives on the criteria most relevant to the potential adopter, among other factors(Rogers, 2003). RA practices with these features are more likely to be adopted by farmers, and thus benefit them. Conversely, RA features are less likely to be adopted or may be dis-adopted later on.

**Economic competitiveness and performance:** In smallholder systems where households depend entirely on their farms for their livelihood, purchasing input to the farm could come at the expense of household subsistence. Thus, the economic competitiveness of an intervention is highly intertwined with its likelihood of being adopted. Economic competitiveness here could relate to the cost of applying the practice relative to the farmers capability to meet those costs (Grabowski et al., 2016; Razafimahatratra et al., 2021) or the opportunity cost of transition. The capacity to meet these costs is linked to performance in terms of yield, ability of the RA practice to reduce crop losses from erratic rain (Grabowski et al., 2016) or pest and diseases (Simtowe & Mausch, 2019) or any parameter most useful to the targeted farmer. It is worth noting that the ability to convert farm outputs (yield) into cash to meet the costs is affected by external forces like access to markets, the various market forces and supporting infrastructure and systems. By addressing the cost factors, optimization of performance of the intervention and diversifying the range of marketable products for instance inclusion of the sale of captured carbon alongside other products (Benjamin et al., 2018), it is possible



to improve the financial outcome of farmers. To obtain the saleable farm products described above
hence experience the performance of the RA intervention, the farmer has to be able to meet the RA
practice requirements such as labour demands (Habanyati et al., 2020), time (Bouwman et al., 2021),
and land (Kurgat et al., 2020). Therefore, a farmers' own resource limitations (Grabowski et al., 2016)
and/or their ability to work around these limitations could be a major limiting factor. Therefore,
interventions that could help bridge such resource gaps for instance improving access to credit could
improve performance.
While mechanisms like persuasion, regulation and incentives have often been used to bridge the
adoption gap for most interventions (Ajayi et al., 2008), positive perception of a RA practice plays a
big role in driving continued adoption. Rogers famously argues in his book 'Diffusion of innovations'
that perceptions come from observing and talking to neighbours who have adopted the
intervention(Rogers, 2003). It is thus important to increasing duration of exposure particularly for
interventions whose benefits could take a long time to get fully realised (Alpizar et al., 2022) while
providing technical support (Habanyati et al., 2020) to address any issues that may emerge during the
exposure period. However, it is important to manage expectations or otherwise risk potential dis-
adoption if the practice does not deliver what it promised (Chinseu et al., 2019). Multi-disciplinary
participatory research and project development processes that integrate farmer knowledge and
experiences could play a big role in matching expectations to the local context and equipping farmers
with the tools and information to effectively apply the RA practice in-order to derive the promised
benefits (Entz et al., 2022; Noordin et al., 2001).
**Cultural and social appropriateness:** Cultural beliefs, norms and traditions shape what is acceptable
and what is not within a given society. In relation to RA adoption, this could relate to; livelihood
strategies for a given group (Agundez et al., 2022)(Agundez et al., 2022), gender roles and associated
resource access rights (Kehinde & Adeyemo, 2017; Khoza et al., 2019; Kunzekweguta et al., 2017;
Ngaiwi et al., 2023)(Kehinde & Adeyemo, 2017; Khoza et al., 2019; Kunzekweguta et al., 2017;
Ngaiwi et al., 2023) and the social-cultural beliefs (myths about certain practices) (Agundez et al.,
2022; Assogbadjo et al., 2012).(Agundez et al., 2022; Assogbadjo et al., 2012). For instance, in areas
of Zimbabwe, pearl millet (*Pennisetum glaucum*) has been promoted as a drought-tolerant alternative
to maize following maize crop failure due to droughts; however, some cultures believe that growing
pearl millet would anger ancestral spirits (Mambondiyani, 2020).(Mambondiyani, 2020). In Northern
Malawi, Bambara groundnuts (*Vigna subterranean*) has been promoted for its high nutritious value,
drought tolerance and soil-enhancing qualities; however, certain groups associate it with death, which
has greatly limited its adoption, distribution and marketing (Forsythe et al., 2015)(Forsythe et al.,
2015). Many of these beliefs associated with particular crops and their uses have a gender element as
well. For instance, while men and youth could support with some agronomic activities in Bambara
groundnut production, it is taboo for them to touch the seed. To improve the tolerance and
acceptability of useful interventions like Bambara groundnuts that could be considered alien in certain
cultural contexts, Moore et al. (2015) suggests intensive education campaigns and extensive sharing
knowledge and new practices through communities of practice, a process they describe as scaling
deep.
As Moore et al. (2022?) suggests, society norms can be moulded and shaped through actions of third-
party entities such as government, intergovernmental and non-government organisations, academia,
faith-based organisations often with competing goals. In the smallholder farming space, one
dimension of competition relevant here is between an approach focused on extending the 'green
revolution in Africa' versus 'scaling RA'. While proponents for each of the possible pathways could
justify their individual investment choices, it is important for the communities whose cultural beliefs,
norms and traditions are at stake to be provided with sufficient information and supported in making
an independent evaluation of their alternatives. In the smallholder setting, this often involves intensive
and consistent agricultural extension, characterised by active farmer participation, practical



demonstration of the RA practice benefits and working with common interest groups. Groups
particularly provide a space for consultation between peers and leverage the power of social influence
towards adoption of group norms (Alexander et al., 2022). In practice, agricultural extension services
and community groups are often affiliated to certain entities whose viewpoints and norms they
champion. Therefore, if one seeks to use existing extension and community structures, it is worth
doing some due diligence on the norms, beliefs and traditions of the organisations overseeing these
structures as well as the individuals implementing them.
**Accessibility** could relate to the intervention itself in case of a product (for example improved seed,
seedlings) or essential inputs in case of a process (for instance, agroforestry, conservation agriculture).
For a product, or process to be considered accessible, it must be available, farmers have to be able to
physically reach the point of supply with ease, and they need to have the rights to use it. Availability
refers to the physical presence of the intended product. In relation to adoption of RA, availability of
land (Kehinde & Adeyemo, 2017; Razafimahatratra et al., 2021), water for irrigation (Maindi et al.,
2020) and essential inputs (Murindangabo et al., 2021) stand out as key determinants. Physical access
on the other hand relates to infrastructural barriers to reaching the point of supply for example poor
road infrastructure (Maindi et al., 2020; Wafula et al., 2016), an isolated geographic location (Abebaw
& Haile, 2013), physical proximity to markets (Abdulai et al., 2021; Kifle et al., 2022; Kunzekweguta
et al., 2017; Mujeyi et al., 2022), and ownership of transport assets (Mujeyi et al., 2022). Rights to use
relate to exclusion of certain groups. The most common example in smallholder context relates to land
tenure (Murindangabo et al., 2021; Owombo & Idumah, 2017; Teklu et al., 2023) and rights to protect
and own trees in agroforestry schemes (Kouassi et al., 2021).
A key aspect in moderating accessibility is information of what is needed, why, where to get it, how to
get it, and so on. It is thus important to ensure that the farmer has access to or know where and how to
access all the essential information associated with the intervention. Awazi et al. (2022) found access
to information, along with access to land and household income as key determinants for choice of
agroforestry system (between no agroforestry, agrosilvipastoral system, silvipastoral system and
agrosilvicultural system) as a climate change adaptation mechanism. The level of access, perception
and trust of any particular information source could vary from group to group thus to effectively
communicate, one has to understand the most favoured sources of information for any particular
group (Djido et al., 2021; Muriith et al., 2021).
Addressing the different dimensions of accessibility calls for often higher-level interventions spanning
from infrastructural projects to policy and market-based interventions. Physical access challenges call
for investments on infrastructure such as roads to improve connectivity and link rural areas to
markets. It also calls for establishment of markets and associated infrastructure closer to the rural
sites. On the other hand, market-based incentives designed to boost supply of these essential inputs
could play an important role in improving and sustaining supply of such essential inputs. Though not
a panacea, enacting appropriate policies to address issues of rights, extensive education, and
enforcement of contracts and agreements could be a possible pathway to addressing issues of rights to
access. While the appropriate solution could vary with the context and nature of the problem, it is
likely that any solution will involve reaching out to different actors at multiple levels of the social-
technological-ecological system. For instance, through enhancement of smallholder groundnut seed,
the Southern Groundnut Platform contributed to 11% increase in area under groundnut cultivation in
Southern Tanzania and resulted in 15% increase in groundnut production between 2012 and 2018
(Akpo et al., 2021). Akpo *et al.* (2021) reports various other cases of multi-stakeholder platforms
improving smallholder seed access in Ghana, Mali, Nigeria, Burkina Faso, Ethiopia, and India.
**Capability:** Capability could be applied to the farmers themselves or to the RA practice being
promoted. When applied to the farmer, capability implies one's ability to effectively apply the RA
practice. Andersson and D'Souza (2014) observed that one of the key limitations to farmers trying out
and adopting conservation farming is the added cost in equipment like the ripper, cost of labour to





gather and apply mulch or control weeds in absence of herbicide. Under these circumstances, access
to affordable credit could provide a viable pathway to improving the capability of smallholders to
apply conservation agriculture practices hence increasing their chances to experience its benefits and
adoption (Kehinde & Adeyemo, 2017; Mujeyi et al., 2022). When it comes the accessing credit from
formal financial institutions, one of the main challenges for smallholders is the limited access to
resources that could serve as security for the credit (Nkonki-Mandleni et al., 2022). Other than
influencing access to credit, access to resources such as land and security of tenure could directly
improve or reduce the capability of the farmer to engage in certain practices. Capability could also
relate to perceived usefulness of the RA intervention, which as Mugandani & Mafongoya (2019) and
Oduniyi & Tekana (2019) observed had a greater influence on adoption than awareness.
When it comes to capability and all the other enabling conditions discussed above, information is key.
In the smallholder context, while multi-media sources such as radios, short term message services on
mobile phones and newsletters could be useful (Oladele et al., 2019), extension service and informal
farmer networks particularly play key roles in information flow (Brown et al., 2017; Djokoto et al.,
2016; Habanyati et al., 2020). Extension here does not limit itself to public extension services (for
examples agricultural officers, forestry officers) but also includes private, and NGO farmer support
services. Beyond facilitating information flow, improvement of perception is favoured by adopting
extension approaches that prioritise farmer participation (Entz et al., 2022) and practical
demonstration of the RA practice benefits (Habanyati et al., 2020). When it comes to farmer networks,
farmers are more likely to choose who to consult based on homophily (people similar to themselves,
e.g., religion, tribe), kinship and/or physical proximity (Giroux et al., 2023). Therefore, strengthen the
social capital in farmer networks, it makes sense to work with groups. Apart from creating rich
information networks and generating peer pressure towards adoption of what the group considers
preferable, groups also provide secondary services that could improve the capability of individual
group members. For instance, cooperatives are formed primarily to support members with among
other services, provision of improved inputs and loans. In-fact, Abebaw & Haile (2013) found that
cooperative members were more likely to possess oxen, have leadership experience and have off-farm
work compared to non-members**.**
**3.0 Reinforcing feedback processes in adoption of RA**
Throughout the various phases during which potential adopters interact with a particular RA practice,
the various aspects of economic competitiveness, accessibility, cultural appropriateness, performance,
and capability interact, influencing the system transition (see Figure 2 below).




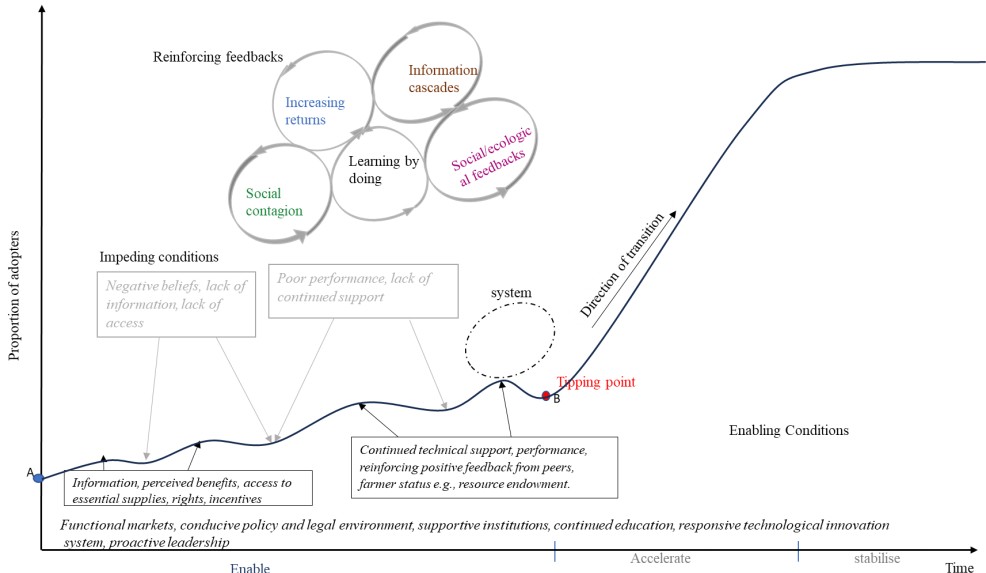


*Figure 2: System transition diagram adapted from Fesenfeld et al. (2022) to show the enabling conditions that are influential at various stages of a farming system transition towards a tipping point. Across the entire transition process, functional markets, conducive policy, and legal environment (e.g., tenure security) coupled with supportive institutions, complementary infrastructure (e.g., roads), continued education to address cultural biases, a responsive technological innovation system (e.g., in terms of capability, functionality and cultural appropriateness) and proactive leadership play a major role.*

In the Enable Phase from when the RA practice is first introduced (point A, Figure 2) to when there is a tipping point of accelerating mass adoption (point B, Figure 2), different factors (enabling conditions) gain importance for different people at different points in time. At the initial stages of introduction, access to information about the practice, perceived benefits of the practice, access to essential supplies and key resources play a key role driving potential adopters to try-out the practice. At the later stages, as people continue interacting with the practice, the performance of the practice, access to continued technical support and feedback from peers gain greater importance in sustaining continued use. As more people use the practice, and demonstrate evidence for its performance, they either attract or discourage others from engaging with the practice, new markets emerge for the products and/or inputs for the practice. At the tipping point (point B, Figure 2), a large enough proportion of the population has adopted the practice such that the rate of adoption becomes self-sustaining and creates further exponential growth in the target population (Lenton et al., 2022; Rogers, 2003). While the factors discussed above independently and in combination enhance the chances of successful adoption at individual and household levels, certain factors independently or in combination could trigger self-propelling, reinforcing feedback processes that could either accelerate or dampen the rate at which the whole community embraces the practice as the norm (scaling out).

Moore *et al.* (2015) describe three possible pathways to scaling of any development intervention; scaling out, scaling deep and scaling up. Scaling out involves impacting greater numbers of people, scaling deep impacting the cultural roots, while scaling up deals with impacting policies and laws. Scaling can occur at an institutional level but is not confined there. Beyond institutional boundaries, processes like scaling deep could influence the culture of an entire community while the influence of



policy and laws in scaling up could extend to other institutional levels including national and
International.

**4.0 A case study of The International Small group and Tree planting programme (TIST) in East**
**Africa.**

TIST is an agroforestry payment for ecosystem service (PES) programme that also promotes
conservation farming (Benjamin et al., 2018). The programme is running in Kenya, Uganda, Tanzania,
and India and over the years, it has reached over 176,000 farming households in 26,996 small groups,
maintained over 22 million trees, and offset over 7 million tonnes of carbon
(https://programme.tist.org). In East Africa, Kenya (15,529 Groups) has the highest number of groups
enrolled followed by Uganda (5,976 groups) (see Figure 3).

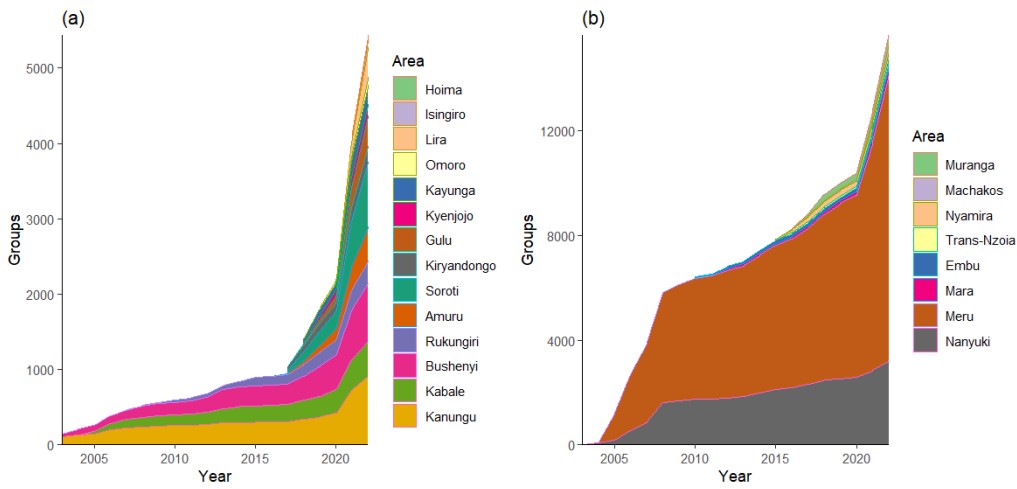

*Figure 3: Enrolment of TIST participant in Uganda (a) and Kenya (b) between 2003 and 2022. The*
*expansion of the programme takes on a different pattern in each of the countries implying that*
*different factors are perhaps involved.*

In Kenya, participant enrolment rates in Meru and Nanyuki overshadow all the other sites in the
country and shape the national enrolment picture while in Uganda, the programme expanded to
several new areas after 2015, with afew (Soroti, Gulu, Amuru and Lira) achieving relatively high rates
of enrolment since introduction of the programme. For instance, of the five sites with the highest
number of groups in Uganda, three sites are less than six years old and among these Soroti has the
second highest enrolment rate of all the sites in the country.

**5.0 Scaling of TIST**

TIST demonstrates all three forms of scaling, scaling out, up and deep (see Table 1).

**6.0 Table 1: Evidence for the different forms of scaling by TIST**

| Scaling type | Evidence for scaling and possible triggers in TIST |
| --- | --- |



| | |
|---|---|
| Scaling-out (mass adoption of TIST practices) | • The number of participants enrolling into TIST in both Uganda and Kenya have continuously increased since initial introduction with 10 out of 18 sites in Uganda enrolling after 2015 (See Figure 3). |
| Scaling-up (TIST adopting good lessons as organisation policy) | • Good practices at group level are shared with other groups in cluster meetings and published in monthly newsletter across all the groups in the country. Through this process, good practices in different locations get integrated across the different project levels and informing programme policy revisions. Through these processes, TIST continuously adapts and changes its policy to deepen and extend its impact.<br><br>• TIST rigorously documents and communicates its impact. Through by doing this, it is influencing changes in design and governance of agroforestry interventions in the region with a number of programmes Kilimanjaro project, itereka and others opting to adapt the TIST model in their implementation as part of the TIST DIY group. |
| Scaling deep (Impacting norms) | • TIST takes deliberate action to ensure that women farmers are represented in groups, constituting at least 40% of group membership composition (Masiga et al., 2012). With group leadership appointed on rotational basis and alternating by gender, women are assured an opportunity to lead the group and access the same trainings and information. The same pattern of alternating leadership occurs at all levels of the programme structure. Through these mechanisms, TIST facilitates gender balance in contexts where such privileges were lacking (Benjamin et al., 2018).<br><br>• TIST conducts routine group trainings on various aspects ranging from financial services, appropriate farming practices and other group relevant aspects to complement the routine extension services provided by the cluster servants. Some of these trainings trigger responses that drive further adoption of the desired practices. For instance, TIST farmers that kept proper records were observed to have more favourable credit compared to those that did not. Proper record keeping was associated to the routine training's farmers received (Benjamin et al., 2018) .<br><br>• Outreach to children of TIST group members who will likely inherit the farms and trees as an opportunity to improve programme stability and sustainability (Masiga et al., 2012). |

**7.0 How is TIST meeting the enabling conditions for enrolment in its sites.**
**Economic competitiveness and performance:** By design, TIST prioritises the minimisation of input
costs while at the same time maximising the benefits from participation in the programme. Being an
agroforestry programme, tree seedlings are an essential input. In the programme, farmers choose
which tree to plant and are encouraged to establish tree nurseries at group level. The localisation of
supply and flexibility of choice potentially improves affordability of seedlings.



TIST further supports its members to access payments for the carbon captured by their trees. These
payments are a supplement to the other benefits farmers already get from planting the same tree
species if they were not in the programme such as soil improvement, erosion prevention, wind breaks,
firewood, fruits from fruit trees, fencing material, timber, medicine, bee habitats, natural insecticides
(Reid & Swiderska, 2008). Benjamin *et al.* (2018) found that women who participated in the TIST
programme were more likely to get a higher profit margin from their agroforestry activities than those
who did not.
Beyond the benefits from agroforestry, participants in TIST also have access to other benefits from
participations like better access to credit(Benjamin et al., 2016), improved social capital, improved
gender equality(Benjamin et al., 2018), livelihood diversification as groups engage in alternative
activities like art and crafts. These various benefits improve the overall performance of the program
and its impact to the lives of those involved.
**Accessibility:** Enrolment into the TIST programme is open to all interested smallholders.
Participation was not restricted by farm size (Benjamin & Blum, 2015) implying that even those with
very small farms could enrol hence increasing accessibility to the programme. Groups establish and
manage their own nurseries which makes seedlings easily accessible by the farmers.
TIST offers farmers contracts of 10-30 years along with regular trainings and extension support in
financial management, tree management and other relevant skills (Masiga et al., 2012). For these
reasons, smallholders in TIST were less likely to be credit constrained and those that kept records
enjoyed more favourable formal credit conditions (Benjamin et al., 2016).
**Cultural appropriateness:** TIST empowers the farmers to make decisions on what is most
appropriate to their contexts for instance. By leaving decisions like what trees to plant, where to plant
them and what group to join to the farmers, the programme ensures that the programme interventions
are appropriate to the farmers context.
TIST farmers are organised in small groups of 6-12 members and 40-50 groups within walking
distance of each other aggregate into a cluster supported by a cluster servant (Masiga et al., 2012).
Farmers in a cluster meet at regular intervals to share good practices, trade experience and share
profits from carbon trade. This localised coordination and knowledge sharing structures creates space
for cultivation of context specific but organisation relevant knowledge, customs, and experience.
**Capability:** TIST does not offer restrictions to various aspects of participation like where to plant
trees hence- increasing the likelihood that many farmers would be capable of participating in the
programme.
TIST trains cluster servants in tree quantification and involves smallholder farmers in the
quantification process hence building their capacity not only understand the processes but also explain
it to others. Hence, empowering them (farmers) not only to access the voluntary carbon
markets(Lenton et al., 2022) but also to support other farmers in the process.
Through the group structure and regular meetings at both the group and cluster level, newly enrolled
participants get to engage with participants who have been in the programme longer. This creates
more opportunities for the farmers to support each other through the adoption process.
**8.0 Reinforcing feedback processes driving adoption of TIST**
Different reinforcing feedback processes are often involved in driving adoption of any given RA
practice. For the case of TIST the processes driving adoption at household and community level could
be summarised into social processes, economic processes, ecological processes, and agronomic
processes as illustrated in the Figure 4 below. The processes often interact at multiple levels,
contributing to yield, income and eventually improved resilience and livelihoods.




*Figure 4:Reinforcing feedback processes driving adoption of TIST at community level. Conservation*
*agriculture and agroforestry improve the soil ecological functioning hence contributing to improved*
*and more stable yields, while the various tree products along with carbon finance contribute to*
*income diversification. Through working in groups, there is better information sharing which in-turn*
*builds and reinforces the social capital. All the various contribute to improved resilience as well as*
*drive social contagion in TIST.*



In some cases, the results of adoption are not always positive, requiring careful analysis of the trade-
offs involved. For instance, Masiga et al. (2012) describes the complex trade-off TIST farmers in
Meru, Kenya have to make in deciding whether to plant eucalyptus (Figure 5). In this case, while the
Green Belt Movement in Kenya discouraged planting of eucalyptus because it could damage the soils
on which they were planted, the Kenya Forest Service promoted eucalyptus for its fast growth to meet
demand for timber and utility poles. Furthermore, Kenyan Power had been vocal about their need for
poles. While the demand for timber and poles could drive more people to plant eucalyptus, its
negative effect on the soil could discourage its adoption.

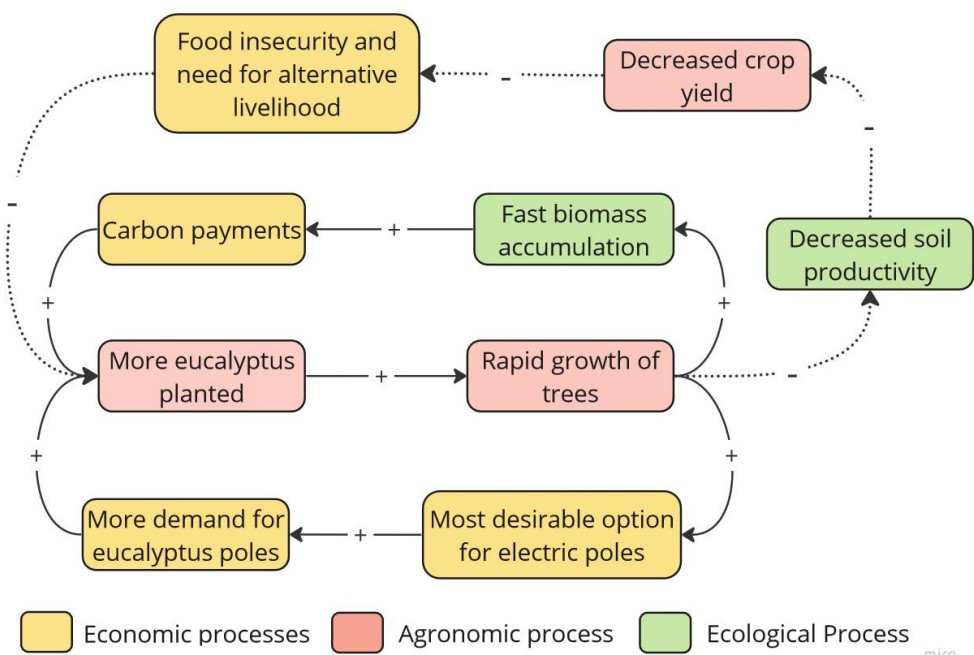


*Figure 5: Reinforcing feedback loops influencing adoption of eucalyptus in Meru, Kenya.*
Apart from reinforcing feedback process that could lead to opposite outcomes like the example above,
some effects are more subtle but equally impactful on adoption. For instance, it has long been
established that gaining information about an initiative precedes adoption (Rogers, 1963). However, if
everyone knew about a practice yet no one has adopted, "*it appears that* the practice has been
deliberately and publicly rejected by everyone" (Centola, 2021, p. 19) hence discouraging other
potential adopters. Various other combinations of factors and actions could lead to different
reinforcing feedback processes with effects that might not be fully predictable. As promoters of
certain interventions, it is worth reflecting on the possible unintended reinforcing feedback processes
triggered by one's actions and taking deliberate steps to strike balance between the factors involved to
increase the chances of achieving the intended system level transition. For instance, to manage the
effect of eucalyptus and its popularity, alongside education about the potential negative effects of
planting eucalyptus, water conserving species such as *Bridelia* and *Sysygium spp* were promoted in
riparian areas through training and additional payments for ecosystem services per indigenous tree
planted within 100 metres of the waterway (Masiga et al., 2012).





While most of our discussion and examples have focused on RA adoption among members of the
same population, well managed reinforcing feedback processes could lead to chain reactions that
drive adoption in populations that are geographically dispersed and also across different levels (see
figure 6). For instance, the positive testimonies from TIST beneficiaries, studies illustrating its
positive impact (see Benjamin et al., 2018; Buxton et al., 2021) and commentaries about its unique
approach to sustainable agro-forestry has made TIST a unique and interesting case both for research
and among development practitioners with various projects like iTeraka in Madagascar, the
Kilimanjaro Project in Tanzania and MyTreesTrust in Zimbabwe adapting different aspects of the
TIST mechanism in their individually unique operations.

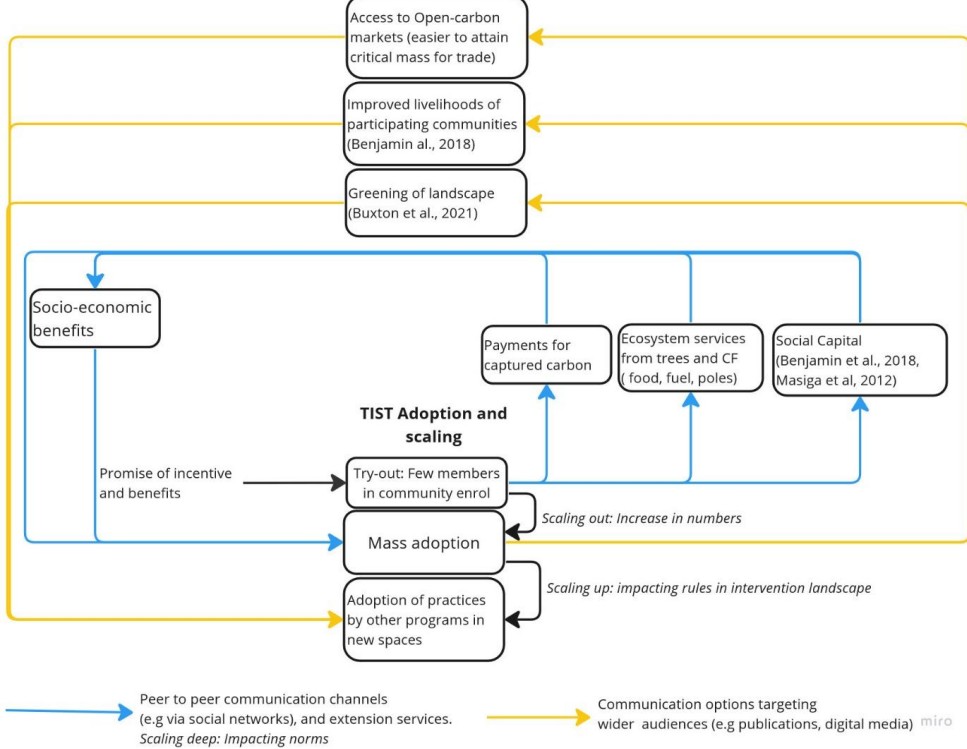


*Figure 6: Reinforcing feedback processes driving multi-level adoption of TIST. Adoption progresses*
*through levels with communication the transition from one level to another.*
Moving from a few individuals trying out the RA practice to a tipping point for mass adoption relies
on a series of multiple peer-to-peer interactions and action and the change occurs at the same level
(community of peers). Success at this level draws attention of stakeholders at different levels or in
different thematic spaces to which the programme lessons could apply, but only if they are
communicated through channels familiar to the independent stakeholder groups. If an interested
stakeholder decides to implement the programme in a new site, then the cycle repeats itself, with new
participants potentially trying out the practice. However, the success in the previous site does not
automatically predict success in a new site, but rather demonstrates the potential if the necessary
enabling conditions can be met or created in the new site.
**9.0 What does the TIST scaling pattern tell us about accelerating RA adoption?**



Most RA practices by their nature offer opportunity to benefit from payments for various
environmental services with such payments potentially reducing the opportunity cost for their
adoption. The successful adoption of TIST is largely attributed to the programme's ability to break the
institutional barriers for farmers to access such payments, allowing them to supplement the numerous
livelihood diversification options and co-benefits offered by agroforestry and CA. In TIST, Farmers
are involved in the monitoring, verification and reporting of the trees carbon content along with
quantifiers in collaboration with international TIST staff (Benjamin et al., 2018). Small groups receive
70% of all the profits from the carbon captured and sold. These profits are shared among group
members in proportion to number of trees each member planted (Masiga et al., 2012).

The growth of TIST largely leverages social capital cultivated and nurtured through participant active
involvement in the programme processes, continued capacity building and working in small groups
with members within walking distance of each other. TIST operates in groups of 6-12 members with
each group required to plant at least 5000 trees over five years depending on availability of land in
order to qualify for payments (Masiga et al., 2012). The social network created by the group structure
facilitates information sharing and support systems that drive adoption (Benjamin et al., 2018) while
the fact that the whole group has a shared tree planting quota, enables distribution of risks and permits
even for farmers with limited access to land to join the programme (Benjamin & Blum, 2015) .

## 10.0    Conclusion

Several studies look into factors that could affect the adoption of various RA farming practices across
sub-Saharan Africa, however, little is still known about what could enable rapid scaling. In this Paper,
we draw on the lessons from the rapid scaling of TIST in East Africa to understand what processes
could be leveraged to rapidly scale other RA interventions in the Global South. We observe that the
successful scaling of TIST could be attributed to: (1) cultivation of social capital through group
structure which enables sharing of risk, facilitates information flow and grows a community of
practice; (2) minimising barriers to farmers directly accessing payments for the carbon captured by
their trees alongside the multiple benefits of agroforestry that they already access. While the subject of
social capital has been relatively well explored in literature, carbon trading is relatively new with
many potential opportunities; such as a catalyst to accelerate adoption of RA practices. A key lesson
other NGOs and programmes can draw from TIST, it is worth thinking about carbon accreditation
processes during RA programme design, the review of ongoing projects and that smallholder farmers
can be an integral part with agency in these processes.

While the data on enrolment of TIST clearly reveals evidence of scaling, it also provokes important
questions on factors and processes responsible for (a) the difference in rates of scaling and (b)
variations in scaling patterns between seemingly similar sites? Finding answers to these questions
could provide insights strategies to address site specific barriers to accelerated adoption. This could be
a potential next step for future research.

**Competing Interests**
The contact author has declared that none of the authors has any competing interests.

**Acknowledgement**
This paper is part of the Oppenheimer Programme in African Landscape Systems (OPALS) jointly
funded by Oppenheimer Generations Research and Conservation, Sarah Turvill and University of
Exeter. The authors thank TIST for providing access to their participant enrolment data that was used
in this paper.



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
