# Peer review of "Positive tipping points for accelerating adoption of regenerative practices in African smallholder farming systems: What drives and sustains adoption?"

_EGUsphere, 2023_

## Author Comment (AC1)

**RC 1 comments and responses.**

The manuscript 'Positive tipping points for accelerating adoption of regenerative practices in African smallholder farming systems: What sustains adoption?' assesses the potential for successful adoption of Regenerative Agriculture in Sub-Saharan Africa. It introduces the concept of positive social tipping processes using existing frameworks and applies them to the example of the International Small group and Tree planting programme (TIST) in East Africa.

While RA adoption and the TIST programme is a very interesting example of a positive social tipping point and the manuscript has the potential to become an interesting publication, the manuscript in its current form lacks clear structure, clear definitions and coherent use of terminology. It requires fundamental reworking before publication.

The structure of the manuscript is not clear. Why do the authors start with the social tipping point framework by FOLU, then use Fesenfeld (2022) and finally move to Moore et al's (2015) concept of scaling? What is the connection between Moore et al and STPs and why is it useful to use them both?

RESPONSE:

Thank you very much for this observation. It is helpful to learn that we needed to do more to ensure that we more clearly communicate our conceptualisation and justification of the choice of concepts and their combination in the manuscript. We explain why these concepts and this combination further:

1. The framework for operationalisation of positive tipping points suggested by Lenton et al. (2022) and illustrated in the FOLU defines tipping elements and how the interactions between these elements could affect system states. Our choice of this illustration was for two main reasons, (1) to introduce the idea of enabling conditions and feedback loops as essential elements in triggering tipping points, (2) give the reader an insight into what constitutes these enabling conditions and feedback processes.

2. We think that fig. 1 in Fesenfield et al. (2022) does a good job in showing the system level effects of the interaction various enabling conditions and reinforcing feedback processes in time, leading to system level transition between states. However, to improve coherence and structural consistency of the manuscript, we choose to replace it with our own illustration, more focused on the conceptual framing used in the manuscript.

3. Moore et al.'s concept of scaling introduces three levels of scaling; scaling deep (impacting social norms), scaling out (impacting greater numbers) and scaling up (Impacting laws and policy). The framework for operationalisation of positive tipping points introduces the concept of reinforcing feedback processes playing a central role in driving the various dimensions and levels of scaling. It crucially observes that, if these processes are strong enough, scaling could be self-perpetuating (Global Tipping Points Report, 2023).

In the revised manuscript version, we shall introduce a section on conceptual framing (just after introduction). In this section we shall explain the relationship between these concepts and use this to set the tone for the rest of the manuscript.

The manuscript is not very well written and requires correction of typos and other language and grammar mistakes before publication. In addition, the manuscript's style (repetition of words, mistakes in referencing) needs to be improved.

RESPONSE:

The authors thank the referee for this critical observation. We shall review the document with the aim of identification and correction of typos, language and grammar mistakes that we could have missed during the earlier reviews.

Section 1:

Fig 1 which is directly copied from the FOLU report is not necessarily suitable to describe a positive social tipping process. Column 3 (conditions for systemic tipping points) is labelled as 'enabling environment' in the FOLU report (non peer-reviewed grey literature) but usually, positive tipping frameworks start with enabling conditions before reinforcing and dampening feedbacks lead to a tipping point. I would recommend using Fig 3 in Lenton et al (2022) or Fig 4.2.3 in the Global Tipping Point Report as framework instead.

RESPONSE:

Thank you referee for this fundamental observation. Figure. 1 was intended to introduce the tipping point elements to the reader. However, as the referee observes, its labelling does not show the sequence of actions in triggering the tipping points. To address this, we

1. introduce a section on conceptual framing to explain the relationship between the two main frameworks used in the manuscript, the positive tipping point framework illustrated in figure 1 and Moore et al. (2015) conceptualisation of scaling.
2. Replace the present figure 1 with a figure illustrating the relationship between these two concepts.

Section 2:

I wouldn't define economic competitiveness as an enabling condition. I would rather define it as social tipping element following Otto et al (2020). An intervention to create an enabling condition to reach economic competitiveness could be investments in R&D or extension services in the RA field. The examples of control variables for enabling conditions provided in the Lenton et al (2022) figure seem more suitable to me.

The categories economic competitiveness, accessibility, capability and cultural appropriateness are neither clearly defined nor coherently applied throughout section 2. For example, access to affordable credit is listed under 'capability'. Extension services are discussed in the competitiveness section. Each category needs to be clearly defined.

The role of information is not clear. Is it an additional category or does it run through the four other categories?

RESPONSE:

Thank you very much for this critical observation. This further highlights the lack of clarity and the inconsistency in definition and application of terminology which the referee has observed elsewhere. For coherence and consistency across the entire document, we adopt the naming in Lenton et al. (2022) in the appropriate places within the document. We also substantially revise the text in these sections to improve clarity and consistency of definitions. The elements under control variables for enabling condition (in the positive tipping points framework) will be merged into the following four categories (price/cost, complementarity and performance, Desirability and symbolism, Accessibility/Convenience, Information, social networks and capability) reflect their interdependencies in regenerative agriculture systems.

Section 3:

Fig 2: Apparently this figure is adapted from Fesenfeld (2022) but the reference is missing in the bibliography and thus, I cannot evaluate it. Also, the four categories (economic competitiveness, accessibility etc) are, according to the text, interacting in Fig 2, but they are not even referenced in Fig 2.

RESPONSE:

Thank you for this critical observation which further highlights the need to review the manuscript and address referencing gaps across the entire document. However, to improve the consistency and coherence across the entire manuscript, we shall not be using this figure any more and substituting it with one that aligns better with our revised structure. Despite these changes, we shall make the appropriate referencing and address any gaps that we could have missed in the earlier review processes.

The section is titled 'reinforcing feedback processes' but they are not discussed in detail in the section. Fig 2 is not well described in the text. It is not clear how Moore's (2015) definition of scaling is linked to the Fesenfeld et al. transition diagram.

RESPONSE:

Thank you very much for this observation. To address this, we

1. Introduce a section on conceptual framing to explain the relationship between the two main frameworks used in the manuscript, the framework for operationalisation of positive tipping points illustrated in figure 1 and Moores et al. (2015) conceptualisation of scaling.
2. Replace the present figure 1 with a more appropriate figure illustrating the relationship between these two concepts.

Section 8:

Why is a causal loop diagram used to describe the positive feedback loops? Who developed it and on what basis? Was it developed together with TIST farmers? Or based on a literature review? This is all very unclear.

RESPONSE:

Feedback loops emerge from a cause-effect relationship. The purpose of this illustration is to easily present these relationships to our potential readers. The causal loop diagram (fig. 4) was developed by the authors based on literature (peer reviewed and grey) on TIST.

Fig 5 is labelled 'reinforcing feedback loops' but shows dampening feedback loops as well. The negative link between 'decreased soil productivity' and 'decreased crop yield' is incorrect. More decreased soil productivity leads to more decreased crop yield. Further, a rapid growth of trees leads to a decrease in soil productivity. The link needs to be positive or the label needs to change to 'soil productivity'. The entire figure needs to be reworked.

RESPONSE:

The authors thank the referee for this critical observation. We have since reworked figure 5 and see the revised figure below

[Figure]

Again, how is Fig 6 linked to the positive social tipping framework? Who has developed the figure, based on what information? Why is the layout different to Figs 5 and 4?

RESPONSE:

Fig 6 was developed by the authors based on literature (peer reviewed and grey) on TIST. The lay-out is different because the focus of this figure is to show transitions across scales rather than a highlighting the internal dynamics that affect these processes which is the focus of fig 4 and 5.

---

## Author Comment (AC2)

Revised Manuscript structure

| Structure before revision | Revised structure |
| --- | --- |
| 1. Abstract
2. Introduction
3. Enabling conditions for successful adoption of RA in Africa
4. Reinforcing feedback processes in adoption of RA
5. A case study of The International Small group and Tree planting programme (TIST) in East Africa
6. Scaling of TIST
7. How is TIST meeting the enabling conditions for enrolment in its sites?
8. Reinforcing feedback processes driving adoption of TIST
9. What does the TIST Scaling pattern tell us about accelerating RA adoption?
10. Conclusion | 1. Abstract
2. Introduction
3. Conceptual framing
4. Enabling conditions and feedback processes for successful adoption of RA in Africa
5. A case study of the International Small Group and Tree planting Programme (TIST) in East Africa
6. Enabling conditions and reinforcing feedback processes in the scaling of TIST
7. What does the TIST scaling pattern tells us about accelerating RA adoption?
8. Conclusion |

---

## Author Response (AR1)

**Contents**

**Responses to reviewer 1 comments**

The manuscript 'Positive tipping points for accelerating adoption of regenerative practices in African smallholder farming systems: What sustains adoption?' assesses the potential for successful adoption of Regenerative Agriculture in Sub-Saharan Africa. It introduces the concept of positive social tipping processes using existing frameworks and applies them to the example of the International Small Group and Tree planting programme (TIST) in East Africa.

While RA adoption and the TIST programme is a very interesting examples of a positive social tipping point and the manuscript has the potential to become an interesting publication, the manuscript in its current form lacks clear structure, clear definitions, and coherent use of terminology. It requires fundamental reworking before publication.

RESPONSE 1:

Thank you for reading our manuscript and providing such useful feedback. We acknowledge the need for substantial changes in the overall manuscript structure, clear definition and coherent use of terminology and concepts, clear referencing, and appropriate signposting of illustrations. We appreciate your suggestions to improve the quality of the manuscript (particularly around the clear structure, definitions, and terminology). To shall take the following steps to address the various observed issues.

1. Introduced a section on conceptual framing (just after introduction). In this section, we discuss the relationship between the positive tipping points framework proposed by Lenton et al. (2022) and Moore et al.'s concept of scaling. We then propose an integration between these two frameworks which forms a basis for the rest of our analysis throughout the manuscript and have ensured that consistent terminology is used accordingly.
2. Replace the original Figure 1 (from the FOLU report) with an adapted version informed by our conceptual synthesis.
3. section 2.0 on enabling conditions will be merged with 3.0 on feedback. Under each of the subheadings in this combined section, we discuss enabling conditions, the amplifying feedback they interact with, and the possible scaling effect. The elements under control variables for enabling condition (in Lenton et al. (2022) positive tipping points framework) have been merged into the following four categories (Cost, performance and capability, desirability and symbolism, Accessibility/Convenience, Information/social networks) reflect their interdependencies in regenerative agriculture systems.
4. The section on scaling of TIST (presently section 5, 6) and feedback processes driving adoption of TIST (section 8) have been merged with the section on enabling conditions driving adoption of TIST (section 7). The section will then cover enabling conditions and amplifying feedback processes in the TIST case study and their impact on scaling. Except for Figure 4 which has been integrated under the section on Cost, performance, and capability, the other figures have been omitted.

We shall now provide more specific responses to each of the comments that follow.

The structure of the manuscript is not clear. Why do the authors start with the social tipping point framework by FOLU, then use Fesenfeld et al. (2022) and finally move to Moore et al.'s (2015) concept of scaling? What is the connection between Moore et al and STPs and why is it useful to use them both?

RESPONSE 2:

Thank you very much for this observation. It is helpful to learn that we needed to do more to ensure that we more clearly communicate our conceptualisation and justification of the choice of concepts and their combination in the manuscript. We explain why these concepts and this combination further:

1. The Tipping Points framework suggested by Lenton et al. (2022) and illustrated in the FOLU defines tipping elements and how the interactions between these elements could affect system states. Our choice of this illustration was for two main reasons, (1) to introduce the idea of enabling conditions and feedback loops as essential elements in triggering tipping points, (2) to give the reader an insight into what constitutes these enabling conditions and feedback processes.
2. We think that figure. 1 in Fesenfield et al. (2022) does a good job of showing the system-level effects of the interaction of various enabling conditions and reinforcing feedback processes in time, leading to system-level transition between states. However, to improve the coherence and structural consistency of the manuscript, we replace it with our own illustration of the interaction between Lenton et al.'s positive tipping points framework and Moore et al.'s dimensions of scaling.
3. Moore et al.'s concept of scaling introduces three levels of scaling; scaling deep (impacting social norms), scaling out (impacting greater numbers), and scaling up (Impacting laws and policy). Changing societal norms and policy plays a crucial role in creating enabling conditions for mass adoption and positive social transformation (Global Tipping Points Report, 2023).

In the revised manuscript version, we introduce a section on conceptual framing (just after introduction). In this section we synthesise the relationship between these concepts and use this to set the tone for the rest of the manuscript.

The manuscript is not very well written and requires correction of typos and other language and grammar mistakes before publication. In addition, the manuscript's style (repetition of words, mistakes in referencing) needs to be improved.

RESPONSE 3:

The authors thank the referee for this critical observation. We shall review the document to identify and correct typos, language, and grammar mistakes that we could have missed during the earlier reviews.

Section 1:

Fig 1 which is directly copied from the FOLU report is not necessarily suitable to describe a positive social tipping process. Column 3 (conditions for systemic tipping points) is labelled as 'enabling environment' in the FOLU report (non peer-reviewed grey literature) but usually, positive tipping frameworks start with enabling conditions before reinforcing and dampening feedbacks lead to a tipping point. I would recommend using Fig 3 in Lenton et al (2022) or Fig 4.2.3 in the Global Tipping Point Report as a framework instead.

RESPONSE 4:

Thank you referee for this fundamental observation. Figure. 1 was intended to introduce the tipping point elements to the reader. However, as the referee observes, its labelling does not show the sequence of actions triggering the tipping points. To address this, we shall

1. introduce a section on conceptual framing to explain the relationship between the two main frameworks used in the manuscript, the positive tipping point framework illustrated in figure 1 and Moore et al. (2015) conceptualisation of scaling.
2. Replace the present figure 1 with a figure illustrating the relationship between these two concepts.

Section 2:

I wouldn't define economic competitiveness as an enabling condition. I would rather define it as a social tipping element following Otto et al (2020). An intervention to create an enabling condition to reach economic competitiveness could be investments in R&D or extension services in the RA field. The examples of control variables for enabling conditions provided in the Lenton et al (2022) figure seem more suitable to me.

The categories of economic competitiveness, accessibility, capability, and cultural appropriateness are neither clearly defined nor coherently applied throughout section 2. For example, access to affordable credit is listed under 'capability'. Extension services are discussed in the competitiveness section. Each category needs to be clearly defined.

The role of information is not clear. Is it an additional category or does it run through the four other categories?

RESPONSE 5:

Thank you very much for this critical observation. This further highlights the lack of clarity and the inconsistency in the definition and application of terminology that the referee has observed elsewhere. For coherence and consistency across the entire document, we shall adopt the naming in Lenton et al. (2022) in the appropriate places within the document. We also substantially revised the text in these sections to improve the clarity and consistency of definitions. The elements under control variables for enabling conditions (in the positive tipping points framework) will be merged into the following four categories (cost, capability and performance, Desirability and symbolism, Accessibility/Convenience, Information, and social networks ) to reflect their interdependencies in regenerative agriculture systems.

Section 3:

Fig 2: Apparently this figure is adapted from Fesenfeld et al. (2022) but the reference is missing in the bibliography and thus, I cannot evaluate it. Also, the four categories (economic competitiveness, accessibility, etc) are, according to the text, interacting in Fig 2, but they are not even referenced in Fig 2.

RESPONSE 6:

Thank you for this critical observation which further highlights the need to review the manuscript and address referencing gaps in the entire document. However, to improve the consistency and coherence across the entire manuscript, we shall not be using this figure any more and substituting it with one that aligns better with our revised structure. Despite these changes, we shall make the appropriate referencing and address any gaps that we could have missed in the earlier review processes.

The section is titled 'reinforcing feedback processes' but they are not discussed in detail in the section. Fig 2 is not well described in the text. It is not clear how Moore's (2015) definition of scaling is linked to the Fesenfeld et al. transition diagram.

RESPONSE 7:

Thank you very much for this observation. To address this, we shall

1. introduce a section on conceptual framing to explain the relationship between the two main frameworks used in the manuscript, the positive tipping point framework illustrated in figure 1 and Moore et al. (2015) conceptualisation of scaling.
2. Replace the present figure 1 with a figure with a more appropriate illustration of the interaction between these two concepts.

Section 8:

Why is a causal loop diagram used to describe the positive feedback loops? Who developed it and on what basis? Was it developed together with TIST farmers? Or based on a literature review? This is all very unclear.

RESPONSE 8:

Feedback loops emerge from a cause-effect relationship. The purpose of this illustration is to easily present these relationships to our potential readers. The causal loop diagram (fig. 4) was developed by the authors based on literature (peer-reviewed and grey) on TIST.

Figure 5 is labeled 'reinforcing feedback loops' but shows dampening feedback loops as well. The negative link between 'decreased soil productivity' and 'decreased crop yield' is incorrect. More decreased soil productivity leads to more decreased crop yield. Further, the rapid growth of trees leads to a decrease in soil productivity. The link needs to be positive or the label needs to change to 'soil productivity'. The entire figure needs to be reworked.

RESPONSE 9:

Thank you so much for this critical observation. To stay focused on the main message and maintain structural alignment, we have had to omit certain elements from the manuscript and merge certain sections. Figure 5 is one of these aspects that will be omitted from the revised manuscript however essential aspects of contents will be integrated into the text.

Again, how is Fig 6 linked to the positive social tipping framework? Who has developed the figure, based on what information? Why is the layout different to Figs 5 and 4?

RESPONSE 10:

Fig 6 was developed by the authors based on literature (peer-reviewed and grey) on TIST.  The layout is different because the focus of this figure is to show transitions across scales rather than highlighting the internal dynamics that affect these processes which is the focus of figure 4 and 5. However, in the revised version, only figure 4 is retained but repositioned.

**Responses to reviewer 2 comments**

The presented manuscript "Positive tipping points for accelerating adoption of regenerative practices in African smallholder farm systems: What sustains adoption?" examines the potential of an accelerated and sustained adoption of regenerative agriculture in Sub-Saharan Africa. It analyses the conditions and feedback processes as concepts of social tipping processes using the example of the International Small Group and Tree Planting Programme, supplemented by a literature review.

The idea of analysing the adoption of RA using a concrete example is interesting and promising, but the approach used is unclear and needs more explanation. For example, it is not clear to me, how the framework for operationalising positive tipping points was used throughout the process and how it was linked to other approaches such as the three forms of scaling by Moore.

RESPONSE 11:

Thank you so much for this critical observation. To explain the relationship between the frameworks, we have introduced a section on conceptual framing (just after the introduction). In this section, we synthesise the relationship between the positive tipping points framework proposed by Lenton et al. (2022) and Moore et al.'s concept of scaling. We then go ahead to propose an integrated framework embracing the relationship between these two frameworks which then forms a basis for the rest of our analysis throughout the manuscript.

Furthermore, the manuscript lacks definitions and a standardised use of terminology. For instance, a clear definition of regenerative agriculture practices is missing. In Section 1, conservation agriculture, climate-smart agriculture, and agroforestry are presented as RA practices. In my understanding, conservation agriculture, climate-smart agriculture and regenerative agriculture are all alternative approaches to conventional agriculture that fall under the umbrella of sustainable agriculture, while RA practices tend to include specific agricultural practices such as reducing tillage or growing cover crops (I would include agroforestry here as well) (e.g. Newton et al., 2020). To avoid confusion, I would suggest giving a clear definition of what is meant by RA practices.

RESPONSE 12:

Thank you for this critical observation and a further call for clarity on definitions. Newton et al. (2020) provides a critical review of the various definitions of regenerative agriculture. Based on their work, they identify two possible definition pathways; process-focused definitions (e.g. reduce tillage, crop rotation, cover cropping) or outcome-focused definitions (e.g. increase carbon sequestration, improve ecosystem health). In this piece, we opted for the outcome-based definition for the following reasons;

1. The description of any system as regenerative is based on its outcomes rather than its components.
2. Different combinations of processes could lead to different outcomes depending on the social-ecological context. Thus, rather than defining the processes (which are rather prescriptive), a focus on the outcomes would permit practitioners to identify the most appropriate processes for their unique contexts.

Therefore, in lines 47-50 we write, "*RA here refers to farming practices that improve soil, water and overall ecosystem health, increase carbon sequestration, increase biodiversity, maintain or improve farm productivity and improve social and economic wellbeing (see Newton et al., 2020)*".

RA being such an important concept in this manuscript, in the revised version, we dedicate paragraph 2 in the introductory section to discuss the definition and practical application of the concept.

Also, it is not clear to me whether the Tree Planting Programme is considered as a practice or a programme that facilitates RA practices; again, a more precise definition would be helpful.

RESPONSE 13:

In Line 280, we describe TIST as "an agroforestry, payment for ecosystem services (PES) programme".

In the revised manuscript, we provide more details if the program operations "*The programme also promotes reforestation, conservation farming and entrepreneurship and operates in small groups of 6-12 farmers (Reid & Swiderska, 2008)*" in the introductory paragraph of the TIST case study.

More detailed comments for the respective sections:

Section 2:

- For me, it is not clear to me where the key factors come from and how they are linked to the framework presented in Section 1. Rogers (2003) is cited for the list of conditions and not the framework for operationalising positive tipping points.
- The structure of the respective paragraphs is not clear to me as well. What should be presented and explained? Description of the conditions (e.g. economic competitiveness) in the context of RA in Africa and measures to create these conditions (e.g. information exchange)? If this is the case, it should be made more explicit.
- In the paragraph about "Cultural and social appropriateness", every citation is double. In addition, the description of the competition to the green revolution in Africa in the second paragraph is not clear to me. What is the green revolution? How does this relate to the condition described?
- In the paragraph about "Accessibility", is it not clear what the different forms of accessibility are? The first sentence is incomprehensible to me in this regard. What is meant by intervention? What is meant by process (the examples given were considered practices in section?)? What is meant by product? In the third part of this paragraph, the references seem to be missing.

RESPONSE 14:

1. Thank you so much for your comments which point out the inconsistency in structure (Both overall manuscript structure and paragraph structure in the section on enabling conditions and amplifying feedbacks). To provide more structural alignment across the entire manuscript, we have introduced a conceptual framing section where we synthesise the relationship between the positive tipping points framework proposed by Lenton et al. (2022) and Moore et al. (2015)'s conceptualisation of scaling.
2. The enabling conditions section (previously section 2.0) has now been merged with the former reinforcing feedback loops section. The structure of paragraphs in this section has then been modified to illustrate the interaction between the enabling conditions, amplifying feedback, and the dimensions of scaling.
3. The 'green revolution' in this context is related to the promotion of external inputs like fertilisers and pesticides with the aim of yield maximisation. It was brought up here to explain the trade-off the farmers have to make and how RA could be given an edge. However, to improve clarity, the use of the word green revolution has been dropped entirely in the re-write.

Section 3:

- The chapter is called "Reinforcing Feedback Processes in adoption in RA", but feedback processes are not mentioned or explained in the text. What do the feedback processes mean for the adoption of RA?
- In Fig. 2, it's not clear how the different conditions from Section 2 are reflected.
- In line 268, a distinction is made between the individual level and the household level. What does this distinction mean with regard to regenerative agriculture? Individual farmers, farming households? Section 8 makes a similar distinction between the household and the community level? I would suggest clearly defining these levels and indicating which levels are of interest or being looked at.

RESPONSE 15:

Thank you for these observations. To set the stage for our discussion, we introduced a conceptual framing section which synthesises the interaction between enabling conditions, feedback processes, and scaling. Recognising the disconnect in our previous presentation of feedback processes, we have merged the discussion of feedbacks with enabling conditions and scaling, and have grounded these interactions with examples from the literature on RA. In this discussion, we also address the various scales and levels of presentation.

Section 5 and 6:

- The table is its own chapter.
- It is not clear to me why the example of TIST is analysed using the three forms of scaling from Moore et al. What is the relationship between the conditions and feedback loops and the three forms of scaling?

RESPONSE 16:

Under the section of Scaling TIST, we present a TIST-focused adaptation of the conceptual framework introduced in the section on conceptual framing. We then follow through with explanation of the enabling conditions and amplifying feedback potentially contributing to the observed scaling pattern of TIST. For consistency and structural coherence, we maintain the terminology and structure in the presentation of this TIST case.

Section 8:

- Figure 4 and 5: It is not clear to me how to read this figure. "Social contagion and network effects" seems to be a category of feedback processes. Do the social, ecological, economic and agronomic processes indicated lead to social contagion? Or does a contagious feedback process result from the interaction of these processes? I would suggest explicitly representing the important feedbacks using causal loop diagrams and labelling indicating the respective feedback processes.
- Small note: In Figure 4 it is feedback processes, in Figure 5 it is feedback loops. I would standardise the descriptions.
- Figure 6: Same comment as for Figure 4 and 5. The figure is difficult to read, a clear indication of the feedback processes would be helpful. To be consistent here, I would suggest also adding the polarities (+/-) as in the other two figures.

RESPONSE 17:

Reflecting on the ineffectiveness of some of these figures in delivering the intended message, we have hence omitted figures 5 and 6 from the revised manuscript and repositioned figure 4. ii

**A summary of changes and modifications in the manuscript revision**

Revision codes

| | |
|---|---|
| Red | Sections absent in revised version.
• Section 8. in submission one has been merged into section 9. in submission two.
• Section 12. in submission one has been merged into section 12. in submission two.
• Section 13. in submission one has been integrated in section 14. in submission two. |
| Green | Added sections to the revised version. |

| Version returned for revision | Revised version |
|---|---|
| 1. Title
2. Authors
3. Author affiliation
4. Corresponding author contacts
5. Abstract
6. Introduction
7. Enabling conditions for successful adoption of RA in Africa
8. Reinforcing feedback processes in adoption of RA
9. A case study of The International Small Group and Tree planting programme (TIST) in East Africa
10. Scaling of TIST
11. How is TIST meeting the enabling conditions for enrolment in its sites?
12. Reinforcing feedback processes driving adoption of TIST
13. What does the TIST Scaling pattern tell us about accelerating RA adoption?
14. Conclusion | 1. Title
2. Authors
3. Author affiliation
4. Corresponding author contacts
5. Author ORCID
6. Abstract
7. Introduction
8. Conceptual framing
9. Enabling conditions and feedback processes for successful adoption of RA in Africa
10. A case study of the International Small Group and Tree Planting Programme (TIST) in East Africa
11. Scaling of TIST
12. Enabling conditions and amplifying feedback processes in the scaling of TIST
13. Conclusion |

Details on corrections in revised version by section.

| Section | Detailed corrections |
|---|---|
| Title | added 'drives and' |
| Authors | No changes |
| Author affiliation | No changes |
| Corresponding author contacts | No changes |
| Author ORCID | Added |
| Abstract | Re-written |
| Introduction | Re-written to align with a new structure |
| Conceptual framing | Added – introduced a conceptual framework on which the rest of the manuscript is based. |
| Enabling conditions and feedback processes for successful adoption of RA in Africa | • Reviewed the labeling of enabling conditions.
• Integrated the discussion of amplifying feedback loops with discussions on enabling conditions |
| A case study of the International Small Group and Tree Planting Programme (TIST) in East Africa | Re-written the introductory paragraph adding more updated information. |

| Scaling of TIST | Presented a conceptual framework customised to TIST (Figure 3) |
|---|---|
| Enabling conditions and amplifying feedback processes in the scaling of TIST | Discussed enabling conditions and amplifying feedbacks in TIST, applying the conceptual framework |
| Conclusion | Drew some key insights and possible future direction. |

---

## Author Response (AR2)

Response to **Reviews** of "**Positive tipping points for accelerating adoption of regenerative practices in African smallholder farming systems: What drives and sustains adoption**."

Dear Editor

We thank you for considering our manuscript and the two reviewers for their positive and constructive feedback. We have made several revisions to the manuscript to incorporate their suggestions and provide our point-by-point responses below.

We have put significant further thought into addressing Reviewer 1's seven remaining concerns. We acknowledge that there had been insufficient clarity and signposting in the various parts. We have revised our manuscript and included new and revised text to improve clarity, better introduce and link the material. Our point-by-point response is provided below.

We thank Reviewer 2 for their positive endorsement of our study and their insightful suggestions to improve the manuscript. We have fully incorporated the suggestions in the revised manuscript and provide a detailed explanation of how we have addressed them below.

In our responses below, the reviewers' comments are in plain black text and our responses in blue text, with quoted additions to the manuscript in italic text. All line numbers refer to the revised manuscript without tracked changes.

Thank you again for your consideration of our manuscript. We look forward to hearing from you.

Your sincerely,

Antony Philip Emenyu

Reviewer 1

Abstract:

'We present three key insights' sounds as if those insights were an outcome of the paper. However, these insights are exactly the outcome of Moore's article. What is the new insight here?

Response 1: Thank you for this critical observation. Since Moore's article is one of our key building blocks, we embrace its observations and build on them in this paper. We have revised our abstract and conclusion to more clearly articulate how we build on Moore's work leading to our new conclusions that extend this discourse. To articulate the conclusion from Moore et al., in lines 7-9 in the abstract we write, "*Moore et al. (2015) contended that scaling up, out and deep is essential for wide scale system change but identified a gap in understanding of how to achieve the three-way scaling goal let alone achieve it quickly*". From lines 12-18 we present our own unique conclusions when we write, "*We present three key insights: (1) it is essential to work with centrally positioned actors capable of and motivated to influence changes in policy and norms towards scaling the intervention such as the smallholder farmers for TIST; (2) these different dimensions of scaling continuously interact, influenced by feedback loops. For sustained scaling it is key to create enabling conditions to trigger reinforcing feedbacks, and; (3) The rate of scaling is a factor of the reinforcing feedbacks at play in a particular location. Therefore, identification of these feedbacks and the appropriate leverage points is key to address location specific scaling challenges thus emphasising the need for context specific data*".

The revised abstract now reads,

"*Mass adoption of regenerative agriculture (RA) practices could improve the resilience and increase productivity of African smallholder farming systems in the face of growing climate change pressures. However, mechanisms to rapidly and sustainably scale-up these RA practices are not yet well understood. Recent research suggests that rapid system transitions towards sustainable practices such as RA can be driven by amplifying feedback loops and if these are sufficiently strong, the system could reach a tipping point of self-propelling change. Moore et al. (2015) contended that scaling up, out and deep is essential for wide scale system change but identified a gap in understanding of how to achieve the three-way scaling goal let alone achieve it quickly. To address this gap, we combine Lenton et al. (2022)'s framework for operationalising positive tipping points with Moore et al. (2015)'s conceptualisation of scaling to understand triggers for rapid scaling in the case of The International Small group and Tree planting programme(TIST) in East Africa. We present three key insights: (1) it is essential to work with centrally positioned actors capable of and motivated to influence changes in policy and norms towards scaling the intervention such as the smallholder farmers for TIST; (2) these different dimensions of scaling continuously interact, influenced by feedback loops. For sustained scaling it is key to create enabling conditions to trigger reinforcing feedbacks, and; (3) The rate of scaling is a factor of the reinforcing feedbacks at play in a particular location. Therefore, identification of these feedbacks and the appropriate leverage points is key to address location specific scaling challenges thus emphasising the need for context specific data*".

The introduction has been clearly improved. Especially the connection between the Moore and the Lenton framework is now clear. However, the positive tipping process by Lenton is not introduced in the revised version. Fig 1 does not really show the elements of a tipping process and a reader without previous knowledge about social tipping will not understand the connection between Fig 1 and Lenton's framework. For example, the connection between enabling conditions and Lenton's framework is not described in the text. Social tipping needs a clear introduction.

Response 2: Thanks for pointing out this observation. We have added the following text in the last introduction paragraph (from lines 72 – 81) to introduce the positive tipping points elements, "*This framework proposes that under certain enabling conditions, some actions can trigger rapid and self-propelling adoption of sustainability innovations driven by reinforcing feedback processes in social-technological or social-ecological systems (Lenton et al., 2022). Economic competitiveness, performance and accessibility of innovations to users, the prevailing cultural and social norms and users' capability can all be key enabling conditions for systemic tipping points, and will vary according to context. Reinforcing feedback processes that may drive scaling of adoption include*

*social contagion, increasing returns to adoption, network effects, information cascades, percolation, co-evolution, ecological positive feedbacks, and social-ecological positive feedbacks. Key intervention areas to strengthen reinforcing feedbacks or create enabling conditions include policy and regulation, private finance and markets, innovation and technology, education and information, behavioural nudges and monitoring and accountability mechanisms”.*

The entire paragraph now reads,

*“In this paper, we draw on the framework for operationalisation of positive tipping points proposed by Lenton et al. (2022) to explore enablers and processes that could accelerate scaling. This framework proposes that under certain enabling conditions, some actions can trigger rapid and self-propelling adoption of sustainability innovations driven by reinforcing feedback processes in social-technological or social-ecological systems (Lenton et al., 2022). Economic competitiveness, performance and accessibility of innovations to users, the prevailing cultural and social norms and users' capability can all be key enabling conditions for systemic tipping points, and will vary according to context. Reinforcing feedback processes that may drive scaling of adoption include social contagion, increasing returns to adoption, network effects, information cascades, percolation, co-evolution, ecological positive feedbacks, and social-ecological positive feedbacks. Key intervention areas to strengthen reinforcing feedbacks or create enabling conditions include policy and regulation, private finance and markets, innovation and technology, education and information, behavioural nudges and monitoring and accountability mechanisms. We combine theories of scaling and the Positive tipping points framework to explore the adoption of RA in sub-Saharan Africa. Specifically, we examine Moore et al.'s three dimensions of scaling to identify the potential role of feedbacks between the spread of adoption between individuals, changes in governance and institutions, and changes in culture, values, and behavioural norms. We draw on literature from various regenerative farming interventions across Africa, using The International Small group and Tree planting programme (TIST) in East Africa as a case study”.*

Figure 1 was design to illustrate the interaction between the reinforcing feedback loops and the dimensions of scaling rather than the various elements in the framework for operationalisation of tipping points. For this reason, we thought introducing the tipping elements in the figure could potentially dilute our intended message.

Amplifying feedback are adopted from Lenton et al (2022) who use the term 'reinforcing feedbacks' and include social contagion, increasing returns to adoption, network effects, information cascades, percolation, co-evolution, ecological positive feedbacks and social-ecological positive feedbacks. Why do the authors here only include social contagion and social-ecological feedback in Fig 1?

Response 3: Social contagion and social-ecological feedbacks were introduced here only as examples of possible reinforcing feedbacks. Since we have now introduced multiple reinforcing feedbacks in the previous paragraph, we have omitted the examples from the diagram to minimise chances of possible miss-interpretation (see the revised Figure 1 below).

[Figure]

*Figure 1. The interaction between the different dimensions of scaling driven by reinforcing feedback processes. Different reinforcing feedback processes can be involved at one time. The reinforcing feedback processes act within and across multiple spatial scales (from local, national to international) and influencing changes to the scaling within and across those levels in the process.*

The section 'enabling conditions' needs an introductory sentence. Now, the different parts seem like a random list of terms and it is unclear that cost, desirability etc are subsections to 'enabling conditions'.

Response: We have taken note and introduced a few sentences (lines 118-128) to introduce the section under the heading "**Enabling conditions and feedback processes for successful adoption of RA in Africa**" in line 114. In lines 115 – 125 we write, "*Enabling conditions are thresholds in system parameters such that small further interventions may trigger rapid, self-propelling change. For example, if an innovation outperforms the incumbent system on key metrics (price, labour costs etc.), adoption is more likely to become self-propelling.  Some of these conditions relate to the innovation itself, such as price and quality. These can be partly addressed at the design stage, but may also be affected further by system dynamics including feedbacks (e.g. prices may be lowered and quality improved through increasing returns to adoption) . Others such as complementarity and performance, desirability and symbolism, accessibility and convenience, information and social networks depend on how the innovation fits within the environment in which it is to be implemented (Lenton et al., 2022). These conditions are highly dynamic, continuously adjusting in response to the actions taken and the feedback processes triggered and modifying the intervention environment. To keep up with these dynamics, implementors have to be highly proactive and adaptive in their response*".*

Regarding response 6: I cannot see any figure that substitutes the original fig. 2.

'Enabling conditions and amplifying feedback processes in the scaling of TIST': The authors only provide details about TIST regarding the enabling conditions that were introduced before. How is this linked to amplifying feedbacks?

Fig 3: The examples the figure provides as 'interventions to amplify feedbacks' are examples of 'enabling conditions', not amplifying/reinforcing feedbacks following Lenton et al (2022).

Response 5: Thanks for the critical observation in Figure 3. We have reviewed the initial heading in

the figure from "*Interventions to amplify feedbacks*" to "*Enabling conditions and linked TIST interventions*". We feel that this also benefits from the introduction of positive tipping elements as you suggested previously. Please see the revised Figure 3 below.

[Figure]

*Figure 3. TIST scales up, deep, and out in multiple ways. The interventions activate and contribute to the amplification of feedback processes that drive scaling out, up and deep and the interaction between them.*

Why is Fig 4 necessary and how does it link to the focus of the study? 'Crop diversification' or 'carbon finance' appear in the figure without any contextualization in the main text.

Response 6: Figure 4 illustrates the various pathways in TIST that lead to accumulation of benefits and encourage continued participation. We have introduced the figure in lines 280-283 when we write, "*The subsequent section then explains the mechanisms through which these enabling conditions result in scaling with Figure 4 illustrating the interconnected and mutually reinforcing membership benefits which have potential to drive strong feedbacks*". We further reference the Figure in line 305 when we write, "*This reduction in cost alongside other ecological and social-ecological reinforcing feedback processes leads to accumulation of benefits thus increasing the returns to participation (See Figure 4)*" (lines 303-305 ).

Response 8 is insufficient. If Fig 4 was developed by the authors, it needs to be better described where, why, with whom etc. From the figure capture I take that it is a reproduction from another study. What's the benefit of reproducing the figure here, especially as it is not properly explained and contextualized in the main text.

Response 7: The section under which Figure 4 is presented discusses how the high benefit to cost ratio in TIST encourages continues participation in the program. Figure 4 illustrates the various pathways through which members could benefit from their participation. In this sense we think the figure adds depth to the discussion in this section, especially given the improved sign posting.

The Figure 4 was reproduced from Figure 4.3.11 in Powell et al. which illustrates the mutually reinforcing feedback processes linked to the benefits of TIST (our specific study case) and has been adapted here to provide similar insights to the reader.

To appropriately signpost figure 4, in the introductory paragraph to the section, lines 280 – 283,  we first introduce the figure when we write, "*The subsequent section then explains the mechanisms through which these enabling conditions result in scaling with Figure 4 illustrating the interconnected and mutually reinforcing membership benefits which have potential to drive strong feedbacks*". From lines 303 – 305 we link the figure in our discussion of the link between reinforcing feedbacks and TIST benefits when we write, "*This reduction in cost alongside other ecological and social-*

*ecological reinforcing feedback processes leads to accumulation of benefits thus increasing the returns to participation (See Figure 4)".*

See figure 4 and caption below.

[Figure]

*Figure 4. Mutually reinforcing benefits evidenced in the literature are likely to strengthen feedbacks and increase the likelihood of further adoption of TIST at community level. Conservation agriculture and agroforestry improve soil ecological functioning and contribute to improved and more stable yields (Rehberger et al., 2023), while the various tree products along with carbon finance contribute to income diversification and improved livelihoods (Benjamin et al., 2018). Through working in groups, there is better information sharing which in-turn builds and reinforces the social capital. Strong and visible benefits to individual farmers or small groups are more likely to feedback on adoption rates through social contagion. Reproduced from Figure 4.3.11 in Powell et al. (2023, p. 43).*

**Reviewer 2**

In the Section "Enabling conditions and feedback processes for successful adoption of RA in Africa" I miss a framing or explanation of the sub categories presented. It is not entirely clear to me where the

categories come from and what decision led to combining some of the categories. I would suggest adding 1 or 2 sentences on this in the introductory paragraph of this section.

Response 8: Thank you for this critical observation and suggestion. We have taken note and introduced a few sentences (lines 115-125) to introduce the section under the heading "**Enabling conditions and feedback processes for successful adoption of RA in Africa**" in line 114. In lines 115 – 125 we write, "*Enabling conditions are thresholds in system parameters such that small further interventions may trigger rapid, self-propelling change. For example, if an innovation outperforms the incumbent system on key metrics (price, labour costs etc.), adoption is more likely to become self-propelling. Some of these conditions relate to the innovation itself, such as price and quality. These can be partly addressed at the design stage, but may also be affected further by system dynamics including feedbacks (e.g. prices may be lowered and quality improved through increasing returns to adoption) . Others such as complementarity and performance, desirability and symbolism, accessibility and convenience, information and social networks depend on how the innovation fits within the environment it is to be implemented (Lenton et al., 2022). These conditions are highly dynamic, continuously adjusting in response to the actions taken and the feedback processes triggered and modifying the intervention environment. To keep up with these dynamics, implementors have to be highly proactive and adaptive in their response*".

In line 136-139 provided a few sentences to explain our logic for combining a few categories and write, "*To realistically illustrate the relational dynamics between some of the contextual factors, we have merged certain enabling conditions in the subsequent discussions. Based on this logic resulting categories include cost, performance and capability, desirability and symbolism, accessibility and convenience, information and social networks*".

The Section "Scaling TIST" consists only of a figure without text. An explanatory text would be helpful here. For example, it is not entirely clear to me how exactly this figure was created and how it is connected to the previous and subsequent sections. In your response to one of my earlier comments, you write, "Under the section of Scaling TIST, we present a TIST-focused adaptation of the conceptual framework introduced in the section on conceptual framing. We then follow through with explanation of the enabling conditions and amplifying feedback potentially contributing to the observed scaling pattern of TIST." It would be nice if a similar description, perhaps even more detailed, could be found in the respective section of the manuscript explaining the points above.

Response 9: Thank you again for this critical observation and suggestion. From line 279-283, just after the heading "Scaling of TIST" (Line 278), we have added a few sentences to explain Figure 3 and also introduce the rest of the sections that follow. From line 279-283 we write, "*Here we apply the conceptual framework introduced to identify key features of TIST's success in scaling. In Figure 3, we adapt Figure 1 to illustrate the enabling conditions specific to TIST. The subsequent section then explains the mechanisms through which these enabling conditions result to scaling with Figure 4 illustrating the reinforcing feedbacks linked to membership benefits*".